

# Effect of dark sweet cherry powder consumption on the gut microbiota, short-chain fatty acids, and biomarkers of gut health in obese db/db mice

Jose F. Garcia-Mazcorro[1,2], Nara N. Lage[3,4], Susanne Mertens-Talcott[4], Stephen Talcott[4], Boon Chew[4], Scot E. Dowd[5], Jorge R. Kawas[6] and Giuliana D. Noratto[4]

[1] Faculty of Veterinary Medicine, Universidad Autónoma de Nuevo León, General Escobedo, Mexico
[2] Research and Development, MNA de Mexico, San Nicolas de los Garza, Mexico
[3] Research Center in Biological Sciences, Federal University of Ouro Preto, Minas Gerais, Brazil
[4] Department of Nutrition and Food Science, Texas A&M University, College Station, TX, United States of America
[5] Molecular Research LP, Shallowater, TX, United States of America
[6] Faculty of Agronomy, Universidad Autónoma de Nuevo León, General Escobedo, Mexico

Corresponding authors
Jose F. Garcia-Mazcorro,
josegarcia_mex@hotmail.com
Giuliana D. Noratto,
gnoratto@exchange.tamu.edu

## ABSTRACT

Cherries are fruits containing fiber and bioactive compounds (e.g., polyphenolics) with the potential of helping patients with diabetes and weight disorders, a phenomenon likely related to changes in the complex host-microbiota milieu. The objective of this study was to investigate the effect of cherry supplementation on the gut bacterial composition, concentrations of caecal short-chain fatty acids (SCFAs) and biomarkers of gut health using an *in vivo* model of obesity. Obese diabetic (db/db) mice received a supplemented diet with 10% cherry powder (supplemented mice, $n = 12$) for 12 weeks; obese ($n = 10$) and lean ($n = 10$) mice served as controls and received a standard diet without cherry. High-throughput sequencing of the 16S rRNA gene and quantitative real-time PCR (qPCR) were used to analyze the gut microbiota; SCFAs and biomarkers of gut health were also measured using standard techniques. According to 16S sequencing, supplemented mice harbored a distinct colonic microbiota characterized by a higher abundance of mucin-degraders (i.e., *Akkermansia*) and fiber-degraders (the S24-7 family) as well as lower abundances of *Lactobacillus* and Enterobacteriaceae. Overall this particular cherry-associated colonic microbiota did not resemble the microbiota in obese or lean controls based on the analysis of weighted and unweighted UniFrac distance metrics. qPCR confirmed some of the results observed in sequencing, thus supporting the notion that cherry supplementation can change the colonic microbiota. Moreover, the SCFAs detected in supplemented mice (caproate, methyl butyrate, propionate, acetate and valerate) exceeded those concentrations detected in obese and lean controls except for butyrate. Despite the changes in microbial composition and SCFAs, most of the assessed biomarkers of inflammation, oxidative stress, and intestinal health in colon tissues and mucosal cells were similar in all obese mice with and without supplementation. This paper shows that dietary supplementation with cherry powder for 12 weeks affects the microbiota and the concentrations of SCFAs in the lower intestinal tract of obese db/db diabetic mice. These effects occurred in

absence of differences in most biomarkers of inflammation and other parameters of gut health. Our study prompts more research into the potential clinical implications of cherry consumption as a dietary supplement in diabetic and obese human patients.

## INTRODUCTION

The digestive tract of humans and other animals has coevolved over millions of years with a complex assemblage of many different types of microorganisms (gut microbiota). This coevolution has brought benefits to both forms of life, with the latter sustaining life in the former by helping regulate digestion of nutrients, behavior and the activity of the immune system (*Conlon & Bird, 2014*). Importantly, some bacterial groups in the gastrointestinal tract have become specialized at surviving upon host-derived compounds (*De Vos, 2017*), while other groups are believed to be more dependent on dietary-derived compounds.

The membership of the gut microbiota (both at the mucus layer and in the lumen) is relatively constant overtime and resilient to change. However, this complex host-microbial ecosystem can also experience extensive variability (both over time within an individual or among different individuals) depending on a variety of factors including the age of the host, dietary patterns, body weight and physical activity. Expectably, diet has strong effects on different aspects of health (*Palmer et al., 2017*; *Dahan, Segal & Shoenfeld, 2017*) and a growing group of researchers have demonstrated that these effects are partly mediated by a change in the composition and/or the metabolic activity of the gut microbiota (*Noratto et al., 2014*; *Garcia-Mazcorro, Mills & Noratto, 2016a*; *Garcia-Mazcorro et al., 2016b*; *Wu et al., 2011*). Although there is still a lot of room for understanding and most studies have only analyzed the fecal microbiota, we now know that dietary modifications can change the composition and activity of the gut microbiota which in turn may promote wellbeing in the host (*Sheflin et al., 2016*; *Velly, Britton & Preidis, 2016*).

Each type of food contain a specific blend of nutrients and other bioactive compounds that can be considered as part of medical strategies to help patients suffering with certain health disorders. Useful examples of these health disorders are obesity, diabetes and associated metabolic conditions (*Via & Mechanick, 2016*), which can be partly treated using foods with more fiber and/or chemical compounds such polyphenols and other anti-oxidants (*Pérez-Jiménez et al., 2010*). These bioactive compounds (i.e., anti-oxidants) have beneficial effects on health and growing evidence suggest that these effects are partly mediated by changes in the gut microbiota (*Noratto et al., 2014*; *Rowland et al., 2017*; *Henning et al., 2017*). The benefits of studying this topic are wide and include a better understanding of mechanisms of action thus widening the potential of certain foods to treat specific health disorders.

Cherries are fruits containing bioactive compounds with beneficial properties on human and animal health. Some well-studied cherries compounds include polyphenolics,
carotenoids, and tocopherols (*Budak, 2017*; *Mikulic-Petkovsek et al., 2016*; *Redondo et al., 2017*). Polyphenolics can influence health because of their anti-oxidative, anti-inflammatory, anti-mutagenic and anti-carcinogenic properties (*Panche, Diwan & Chandra, 2016*). Despite the evidence in laboratory animals showing a potential of cherries to help patients with weight disorders (*Wu et al., 2014*; *Song et al., 2016*; *Wu et al., 2016*), and the likely involvement of the gut microbiota in this phenomenon (*Faria et al., 2014*), to our knowledge there are no studies that have investigated the effect of cherry consumption on the gut microbiota using obesity models. Therefore, the objective of this study was to investigate the effect of cherry supplementation on the gut microbiota, SCFAs and biomarkers of gut health using an *in vivo* rodent model of obesity.

## MATERIAL AND METHODS

### Study design

The experimental analyses carried out in this manuscript were approved by the Institutional Animal Care and Use Committee at Texas A&M University (IACUC 2013-0149). Two diets were utilized in this study, one with and one without supplementation with dark sweet cherry (Prunus avium) powder (Table 1 and Table S1). Both diets were adjusted to contain the same amount of energy (Table 1). Leptin receptor-deficient obese db/db mice (BKS.Cg-+Lepr$^{db}$/+Lepr$^{db}$/OlaHsd—fat, black, homozygous) received a diet without cherry supplementation (obese control, $n = 15$) and with 10% cherry powder supplementation ($n = 15$). Lean mice (BKS.Cg-Dock$^{7m}$+/+Lepr$^{db}$/OlaHsd—lean, black, heterozygous, $n = 10$) were used as lean controls and fed a standard diet (i.e., without cherry supplementation, Table 1). All mice were purchased from Envigo RMS, Inc. (Houston, TX, USA). Agar based diets were prepared with AIN-93G diet ingredients as reported in detail elsewhere (*Noratto, Chew & Ivanov, 2016*). The election of an agar-based diet allowed fulfilling the food and part of the water requirement of mice and preserved bioactive compounds in cherry because of the physical properties of agar that remains liquid at $40-45\,°C$. Food and water were provided ad libitum every day for 12 weeks. Food intake and waste were daily recorded. Body weight was recorded once a week and body mass indexes (BMIs) were calculated by dividing body weight (kg) by body length (m$^2$) at the end of the study (*Jeyakumar, Vajreswari & Giridharan, 2006*).

### Blood and tissue collection

Mice were terminated at week 12 after ~12 h fasting by gradual exposure to $CO_2$ inhalation until the animal became unconscious, followed by cervical dislocation. Blood obtained by cardiac puncture was collected into a tube containing 10 µL of heparin and centrifuged at 10,000 rpm at $4\,°C$ to obtain blood plasma. Blood plasma samples were aliquoted and frozen at $-80\,°C$ until analysis. The transverse colon was removed and divided into three sections; one section was fixed in 10% neutral formalin buffer overnight and maintained in 70% ethanol at $4\,°C$ for histological analysis of wall layer thickness (see below) while the other two sections were cleaned from intestinal content with 100 mM phosphate buffer solution (PBS, pH = 7) and either kept in RNA later$^®$ (Applied Biosystems, Foster City, CA, USA)

**Table 1** Diets utilized in this study without cherry supplementation (control) and with 10% cherry supplementation.

| Ingredient | Control | | Cherry 10% | |
| --- | --- | --- | --- | --- |
| | Weight (g) | Kcal | Weight (g) | Kcal |
| Casein | 100 | 400 | 100 | 400 |
| Maltodextrin | 66 | 264 | 66 | 264 |
| Sucrose | 50 | 200 | 50 | 200 |
| Cellulose | 25 | 0 | 25 | 0 |
| Mineral Mix[a] | 17.5 | 0 | 17.5 | 0 |
| Vitamin Mix[b] | 5 | 0 | 5 | 0 |
| L-Cysteine | 1.5 | 6 | 1.5 | 6 |
| Choline Bitartrate | 1.25 | 0 | 1.25 | 0 |
| t-Butylhydroquinone | 0.007 | 0 | 0.007 | 0 |
| Cornstarch | 198.75 | 795 | 98.75 | 395 |
| Soybean Oil | 35 | 315 | 35 | 315 |
| Cherry powder[c] | 0 | 0 | 100 | 400 |
| *Agar* | *20* | *0* | *20* | *0* |
| *Water* | *480* | *0* | *480* | *0* |
| **Total** | **1,000** | **1,980** | **1,000** | **1,980** |

Notes.

[a] AIN-93G-MX supplied by Dyets Inc. (Bethlehem, PA, USA), containing (g/kg): Calcium Carbonate (357), Potassium Phosphate, monobasic (196), Potassium Citrate .H20 (70.78), Sodium Chloride (74), Potassium Sulfate (46.6), Magnesium Oxide (24), Ferric Citrate, U.S.P. (6.06), Zinc Carbonate (1.65), Manganous Carbonate (0.63), Cupric Carbonate (0.3), Potassium Iodate (0.01), Sodium Selenate (0.01025), Ammonium Paramolybdate .4H20 (0.00795), Sodium Metasilicate .9H20 (1.45), Chromium Potassium Sulfate .12H20 (0.275), Lithium Chloride (0.0174), Boric Acid (0.0815), Sodium Fluoride (0.0635), Nickel Carbonate (0.0318), Ammonium Vanadate (0.0066), Sucrose, finely powdered (221.026).

[b] AIN-93G Vitamin Mix supplied by Dyets Inc. (Bethlehem, PA, USA), containing (g/kg): Niacin (3), Calcium Pantothenate (1.6), Pyridoxine HCl (0.7), Thiamine HCl (0.6), Riboflavin (0.6), Folic Acid (0.2), Biotin (0.02), Vitamin E Acetate (500 IU/g) (15), Vitamin B12 (0.1%) (2.5), Vitamin A Palmitate (500,000 IU/g) (0.8), Vitamin D3 (400,000 IU/g) (0.25), Vitamin K1/Dextrose Mix (10 mg/g) (7.5), Sucrose (967.23).

[c] Cherry powder contributed with 5.1 g fiber/kg diet and 759 mg GAE/100 g of total phenolics (629 mg GAE/100 g extractable and 130 mg GAE/100 g non-extractable or bound phenolics). Cherry powder was processed by Powder Pure (The Dalles, OR, USA) and contains 80% dark sweet cherry puree (Bing variety), 20% organic rice maltodextrin and 2% silicon dioxide.

or subjected to scraping the mucosa off and kept in RNA later® (named colonic mucosal cells). Samples maintained in RNA later® were stored at −80 °C for further analysis.

## Bioactive compounds in cherry powder

Dietary fiber was quantified by Retch laboratories (Arden Hills, MN, USA) following standard analytical protocols. Total extractable phenolics were extracted as previously reported (*Condezo-Hoyos, Mohanty & Noratto, 2014*). Briefly, cherry powder (0.5 g) was homogenized with 3 mL acidic methanol (HCl)/water solution (50:50 v/v, pH 2) and left for 1 h at room temperature and constant shaking, followed by centrifugation at 4,000× g for 10 min at 4 °C to obtain the acid methanolic extract in the supernatant. The precipitate was extracted with 3 mL of acetone/water solution (70:30 v/v) (1:5 ratio, v/v) by agitation for 1 h at room temperature, followed by centrifugation at 4,000× g for 10 min at 4 °C to obtain the acetone extract in the supernatant. The combined supernatants were analyzed for total extractable phenolics by Folin Ciocalteu method (*Condezo-Hoyos, Mohanty & Noratto, 2014*), using a standard curve of gallic acid (0 to 0.2 mg/mL) and

expressed as gallic acid equivalents (GAE). The residues were subjected to alkali treatment for extraction of non-extractable or bound phenolics as reported (*Luo et al., 2016*). Briefly, 3 mL of NaOH (4 M) were added to residues after extractable phenolics were recovered and maintained in agitation for 2 h under nitrogen atmosphere in a screw capped vial, followed by centrifugation at $4,000\times$ g for 10 min at 4 °C. The recovered supernatant was adjusted to pH 2 with HCl (6 M) and analyzed for non-extractable bound phenolics using the Folin Ciocalteu method (*Condezo-Hoyos, Mohanty & Noratto, 2014*).

## DNA extraction

Colon content and colonic mucosal samples scrapped from terminal colon were collected from all mice at the end of the study and used to purify total genomic DNA using a commercial DNA extraction kit (Zymo Research Corp, Irvine, CA, USA). DNA samples were adjusted to 5 ng/μL and used for two different analyses (high-throughput 16S sequencing and qPCR analyses).

## High-throughput 16S sequencing for colonic microbiota

DNA samples extracted from terminal colon contents were used to amplify a small (∼300 bp) fragment of the 16S rRNA gene using the primers F515 (5′–GTGCCAGCMGCCGCGGTAA–3′) and R806 (5′–GGACTACHVGGGTWTCTAAT–3′) for further high-throughput sequencing as shown elsewhere (*Garcia-Mazcorro, Mills & Noratto, 2016a*). PCR reactions and 16S sequencing were performed at the Molecular Research LP (MRDNA, Shallowater, Texas USA). The MiSeq instrument (Illumina) was used for sequencing the 16S amplicons following the manufacturer's instructions at MRDNA. This technology has been used in several studies and is recommended by the Earth Microbiome Project (*Caporaso et al., 2012*). Raw 16S data was obtained from MRDNA and analyzed using the freely available bioinformatics pipeline QIIME v.1.8 with default parameters. MRDNA conveniently provides users with files containing joined reads (full.fasta and full.qual files). These files were combined in one single fastq file using QIIME. The resulting fastq file was then used to split sample libraries accordingly to the 8 nucleotide barcodes using the split_libraries_fastq.py in QIIME. Operational Taxonomic Units (OTUs) are operational definitions used to classify 16S rRNA gene sequences from related and unrelated microorganisms and there is debate regarding the best approach to select OTUs from 16S sequences (*He et al., 2015*). In this study we used two approaches to select OTUs. First, we used an open reference algorithm (*Rideout et al., 2014*), which has the advantage of not discarding sequences that do not match the sequence database. The OTU table generated by this approach was used for all diversity and taxonomic analyses. Second, we used a closed reference approach where sequences are discarded if they do not have a close match with the reference sequences. The OTU table generated using this closed approach was used for predicting functional profiles using PICRUSt (see Prediction of metabolic profile below). In this study we used the v. 13_5 of the GreenGenes OTU representative 16S rRNA sequences as the reference sequence collection (*DeSantis et al., 2006*). The phylogenetic method UniFrac (Unique Fraction metric, *Lozupone & Knight, 2005*) was used to investigate differences in microbial communities. Please note that it

is important to investigate both quantitative (weighted) and qualitative (unweighted) UniFrac diversity measures because they can lead to different insights into the factors responsible for structuring microbial communities as shown elsewhere (*Lozupone et al., 2007*). All sequence data and associated metadata was uploaded into the Sequence Read Archive at NCBI (SRP117747).

## qPCR for colonic microbiota and for colonic mucosal samples

Unlike high-throughput sequencing (which in this study was only used to analyze the microbiota in colon contents), DNA samples from both colon contents and from colon mucosal samples were used to perform quantitative real-time PCR (qPCR) using primers targeting the 16S rRNA genes for specific groups of microorganisms (Table S2) based on the Gut Low-Density Array (GULDA, *Bergström et al., 2012*) approach and other publications (*Garcia-Mazcorro et al., 2012*; *Noratto et al., 2014*; *Yang et al., 2015*). All qPCR reactions were carried out at Texas A&M University using the described methodology by *Bergström et al. (2012)* with modifications. Briefly, in this study all assays were ran using a standard curve and these standard curves were constructed using different concentrations of DNA from either the specific microorganisms (*Bacteroides fragilis*, *Lactobacillus plantarum* NRRL No B-4496, *E. coli* NRRL No B-766) or from samples containing high amounts of the desired organism (e.g., a standard curve for Ruminococcaceae was constructed using serial dilutions of a sample with high amounts of Ruminococcaceae DNA as determined by qPCR). DNA samples were adjusted to 5 ng/µL. qPCR data is expressed as log amount of DNA (picograms of amplified DNA) for each bacterial group per 10 ng of total DNA (*Bell et al., 2014*).

## Prediction of metabolic profile

Phylogenetic investigation of communities by reconstruction of unobserved states (PICRUSt, *Langille et al., 2013*) was used to predict the metabolic profile based on 16S sequencing data. For this analysis, we used the OTU table obtained from the closed reference approach described above. PICRUSt results were visualized and analyzed using STAMP (*Parks & Beiko, 2010*) with default parameters. PICRUSt analysis was performed using the OTU table containing all taxa (full OTU table) and also using filtered OTU tables containing a subset of taxa to explore contributions of different taxa separately.

## Short-chain fatty acids (SCFA) analysis

Caecal contents were homogenized with MilliQ water in a proportion of 1:1.5 (weight:volume) and centrifuged at 12,000 g for 10 min. Supernatants were then filtered through a 0.45 µm Nylon filter (VWR® Syringe Filters; VWR, Houston, TX, USA) and analyzed by high-performance liquid chromatography (HPLC) as reported in detail elsewhere (*Campos et al., 2012*; *Garcia-Mazcorro, Mills & Noratto, 2016a*). Butyric acid, methyl-butyric acid, caproic acid, sodium acetate, sodium propionate, and valeric acid were purchased from VWR and used as standards to quantify their caecal contents based on retention time and area of peaks at $\lambda = 220$ nm.

## Histological analyses of colon tissue sections

Paraffin-embedded colon tissues were transversally cut (5 μm thickness) and stained with H&E for microscopic analysis. The thickness of outer colon wall layer was calculated in ImageJ (http://rsb.info.nih.gov/ij/) using 10 measurements (ratio of outer colon wall area to total (outer and inner) colon wall area) from each individual mouse. Photomicrographs were taken with Aperio CS2 digital pathology scanner (Leica Biosystems Inc, Buffalo Grove, IL, USA) and blinded analyzed with regards to treatment group.

## Endotoxin levels in caecal contents and plasma

Caecal contents and blood plasma were subjected to endotoxin analysis using the Endpoint Chromogenic LAL Assay following the manufacturer's protocol (Lonza Walkersville, Inc., Walkersville, MD, USA). Briefly, caecal contents were weighted, suspended in milliQ water (1:1.5, w:v), centrifuged at 12,000 g for 10 min and supernatants transferred to a glass vial for endotoxins quantification. Endotoxin units (EU) were calculated as EU/mg caecal content.

## mRNA levels in colonic tissue and mucosal cells

Biomarkers of inflammation, cellular stress, and gut barrier function were analyzed in colonic tissue and mucosal cells. Briefly, tissues or scrapped mucosal cells were mechanically pulverized in liquid nitrogen. RNA was extracted using TRIzol® LS Reagent (Life technologies, Carlsbad, CA, USA) according to the manufacturer's protocol. Purification was carried out with Direct-zol™ RNA MiniPrep (Zymo Research Corp, Irvine, CA, USA) according to the manufacturer's protocol. Quantification of mRNA was performed using the ND-1000 spectrophotometer (Nanodrop Technologies, Rockland, DE, USA). Purified mRNA was used to synthesize cDNA using iScript™ cDNA Synthesis Kit (BioRad, Hercules, CA, USA). Quantitative real-time polymerase chain reaction (qRT-PCR) was carried out with the SsoAdvanced™ Universal SYBR® Green Supermix (BioRad, Hercules, CA, USA) on a CFX384 Touch Real-Time PCR Detection System (BioRad, Hercules, CA, USA). The reaction volume was 10 μL and all primers were used at a final concentration of 100 nmol/L. The RT-PCR data was analyzed by the 2-$\Delta\Delta$CT method in reference to ribosomal protein L19 (RPL19) as housekeeping gene (*Schmittgen & Livak, 2008*). Primers were purchased from Integrated DNA Technologies, Inc. (San Diego, CA, USA; Table S3). Product specificity was examined by dissociation curve analysis.

## Statistical analysis

Relative abundances of taxa based on sequencing data, mRNA expression, and qPCR data were compared using the non-parametric Kruskal–Wallis test and multiple comparisons were adjusted using Bonferroni in PAST (*Hammer, Harper & Ryan, 2001*). PAST was also used to perform Principal Coordinate Analysis (PCoA) using the weighted and unweighted UniFrac distance matrices obtained from QIIME. The Kruskal-Wallis test was also used for comparison of predicted functional features in STAMP (*Parks & Beiko, 2010*). The non-parametric ANOSIM and Adonis tests were performed for determining whether the grouping of samples by a given category is statistically significant in QIIME. Spearman's correlations matrices featuring data from sequencing analyses, SCFAs, and mRNA levels

in colonic mucosal cells identified by Kruskal-Wallis test as significant ($p < 0.05$), were performed using R studio 3.4.0. SCFAs were compared using the Mann Whitney test when comparing only two treatment groups due to lack of detectable values in one group.

## RESULTS

Cherry powder contributed with phenolics and dietary fiber as bioactive compounds that might reach the lower intestinal tract because of their low bioavailability and most likely modulate microbial populations in the large intestine. Cherry powder had 5.1% fiber and 759 mg GAE/100 g of total phenolics (Table 1). A recent study has thoroughly analyzed the profile of phenolics in dark sweet cherry varieties using mass spectrometry and reported approximately 86 compounds including phenolics, anthocyanins, flavan-3-ols and flavonols (*Martini, Conte & Tagliazucchi, 2017*).

### Host physiology

Several obese animals died for reasons unrelated to the study (five animals from obese control group, three animals from cherry group), all other mice remained visually healthy throughout the study. Body weight, BMIs, the percentage of adiposity, epidydimal and mesenteric fat as well as liver weight were similar in all obese mice (with and without cherry supplementation) and significantly higher compared to lean controls (Table S4). The weight of cecum contents was significantly higher in cherry supplemented mice (314 mg, 198–439 mg, median and interquartile ranges respectively) compared to lean (128 mg, 93–152 mg) and obese controls (191 mg, 104–234 mg) ($p = 0.003$, Table S4), in part reflecting the higher amount of fiber in the cherry-supplemented diet (Table 1).

### High-throughput 16S sequencing for colonic microbiota

High-throughput 16S sequencing allows a deep analysis of complex microbial communities such as the gut microbiota. In this study, the split libraries script yielded a total of 3,171,568 good-quality 16S sequences for analysis ($n = 32$ across all treatment groups, median sequence length: 300 nucleotides). The number of sequences per sample varied from 61,284 (lowest) to 142,829 (highest). All analyses were performed using a rarefaction depth of 61,000 sequences per sample.

Overall the colonic microbiota was dominated by six main taxa at the order level representing four phyla: Bacteroidales (phylum Bacteroidetes), Clostridiales and Lactobacillales (phylum Firmicutes), Verrucomicrobiales (phylum Verrucomicrobia), Desulfovibrionales and Enterobacteriales (phylum Proteobacteria). Together, these taxa comprised about 20 different bacterial families which accounted for over 95% of all sequences in most samples (Fig. 1). The results of sequencing and/or qPCR showed significant differences in the relative abundance of several members of all these main taxa in supplemented mice.

The phylum Bacteroidetes contains several bacterial groups associated with health and this group is usually highly abundant in feces and intestinal contents of human and laboratory mice (*Karlsson et al., 2010*). The family S24-7 (a group of fiber degraders) was very similar in lean (median = 19.8%) and cherry-supplemented (median = 20.3%) and

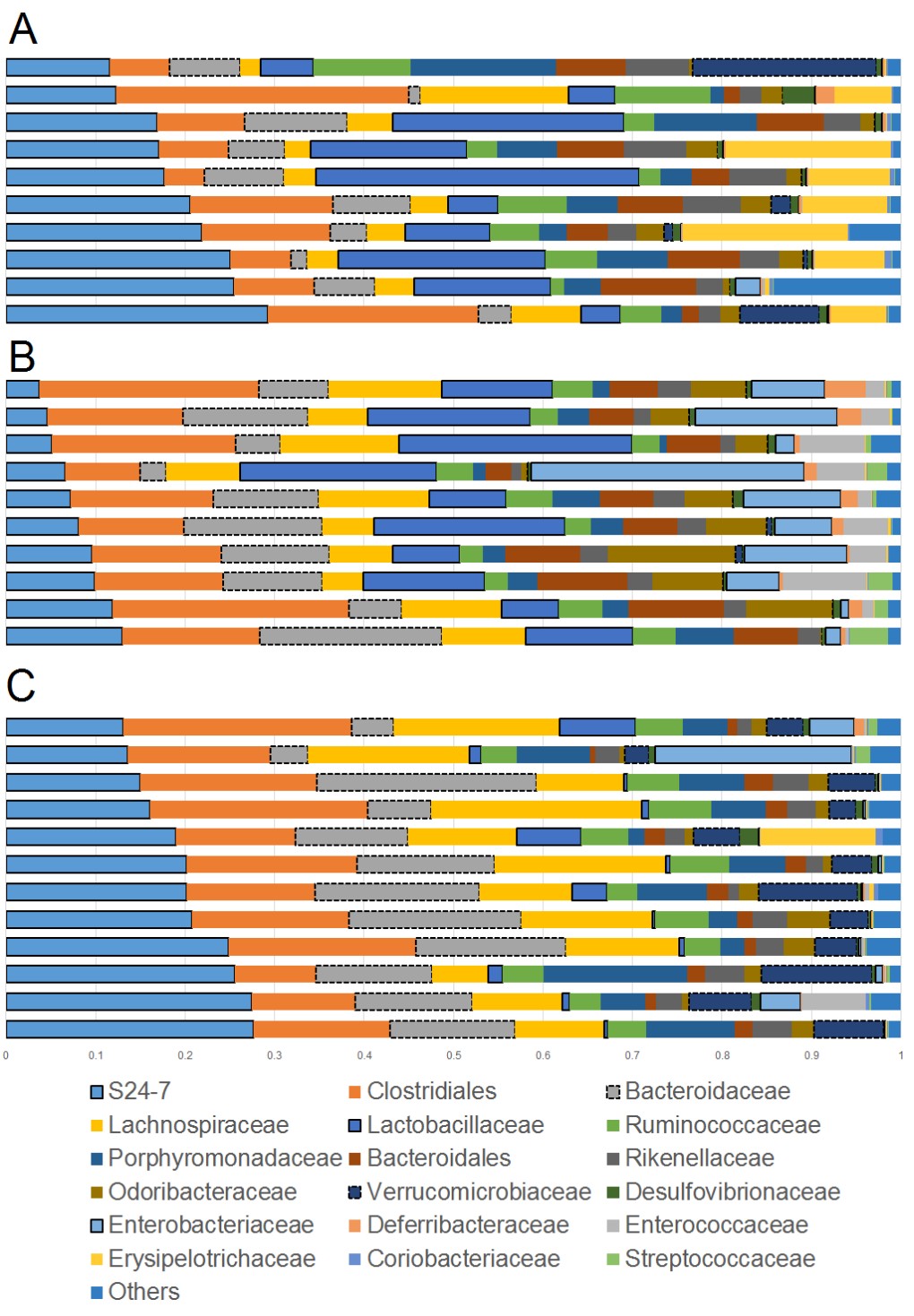

**Figure 1  Bar plots showing relative abundance (percentages, *x* axis) of the most abundant bacterial taxa at the family level.** (A) Lean; (B) Obese; (C) Cherry supplemented group. Please note the noticeable difference in the abundance of the S24-7 group, Bacteroidaceae, Lactobacillaceae, *Akkermansia* (family Verrucomicrobiaceae), and Enterobacteriaceae (highlighted for better visualization). Statistical significant differences were found for these groups using either 16S sequencing, qPCR analyses, or both (see main text for details).

both were about 3 times higher when compared to obese mice (median = 7.9%, $p < 0.005$ for both comparisons) (Fig. 1). While this result supports the observed higher weight of cecum contents in supplemented mice, it also opens up the question of why lean controls (not supplemented) also showed similar levels of S24-7 compared to supplemented mice. The family Bacteroidaceae was similar in all obese mice but only supplemented mice had higher abundance (median = 13.5%) compared to lean controls (median = 6.6%, $p = 0.0124$). This is interesting because *Bacteroides* has been linked to production of SCFAs (*Chen et al., 2017*; *Rios-Covian et al., 2017*) and at least one group of *Bacteroides* (*B. acidifaciens*) has been shown to also use host compounds (*Berry et al., 2013*; *Sonnenburg et al., 2005*) similarly to *Akkermansia*.

The phylum Firmicutes is also a highly abundant member of the gut microbiota and contains many groups associated with health, for example producers of SCFAs (*Barcenilla et al., 2000*). Interestingly, we did not find any difference in the abundance of the two most abundant families within the Firmicutes: Ruminococcacea and Clostridiaceae. On the other hand, *Lactobacillus* was much lower (~10 times lower) in cherry-supplemented mice (median = 0.8%) compared to both lean (median = 12.4%) and obese (median = 13%) controls ($p < 0.005$ for both comparisons), a finding that was also noticeable at the family level (Fig. 1) and that was confirmed using qPCR (see qPCR below).

The phylum Verrucomicrobia is usually low in abundance in the lower gut but it also contains important bacterial groups that have been associated with health such as the mucin-degrader *Akkermansia* (*Derrien et al., 2008*). In this study, the genus *Akkermansia* (family Verrucomicrobiaceae) was lower in obese mice (median = 0.07%) compared to lean (median = 0.31%, $p = 0.0402$) but especially to cherry-supplemented mice (median = 4.9%, $p = 0.0003$), a result that was also confirmed by qPCR. This result was also confirmed using qPCR (see qPCR below).

The phylum Proteobacteria (main order Enterobacteriales) contains bacteria that are usually associated with harmful effects on intestinal health such as several strains of *Escherichia* and *Salmonella*. In this study, the family Enterobacteriacea was more similar between lean (median = 0.08%) and cherry-supplemented (median = 0.3%) mice and both were much lower compared to obese mice (median = 7.3%, $p < 0.05$ for both comparisons, Fig. 1), a finding that could be considered a positive effect of cherry supplementation. The lower abundance of Enterobacteriaceae in supplemented mice was also confirmed using qPCR (see qPCR below). In this study Enterobacteriaceae was the only family within the order Enterobacteriales but most sequences belonged to an unknown genus. The family Alcaligenaceae (Betaproteobacteria) was higher in supplemented mice (median = 1.1%) compared to both lean (median = 0.06%) but especially to obese (median = 0.01%) controls ($p < 0.005$ for both comparisons). There was no significant difference in the abundance of Desulfovibrionales (class Deltaproteobacteria).

Finally, the family Bifidobacteriaceae (order Coriobacteriales, phylum Actinobacteria) was not detected at all with our sequencing effort. Nonetheless, supplemented mice showed similar abundance of the order Coriobacteriales (median = 0.1%) compared to lean (median = 0.3%) but only lean mice was higher compared to obese (median = 0.08%, $p < 0.0001$).

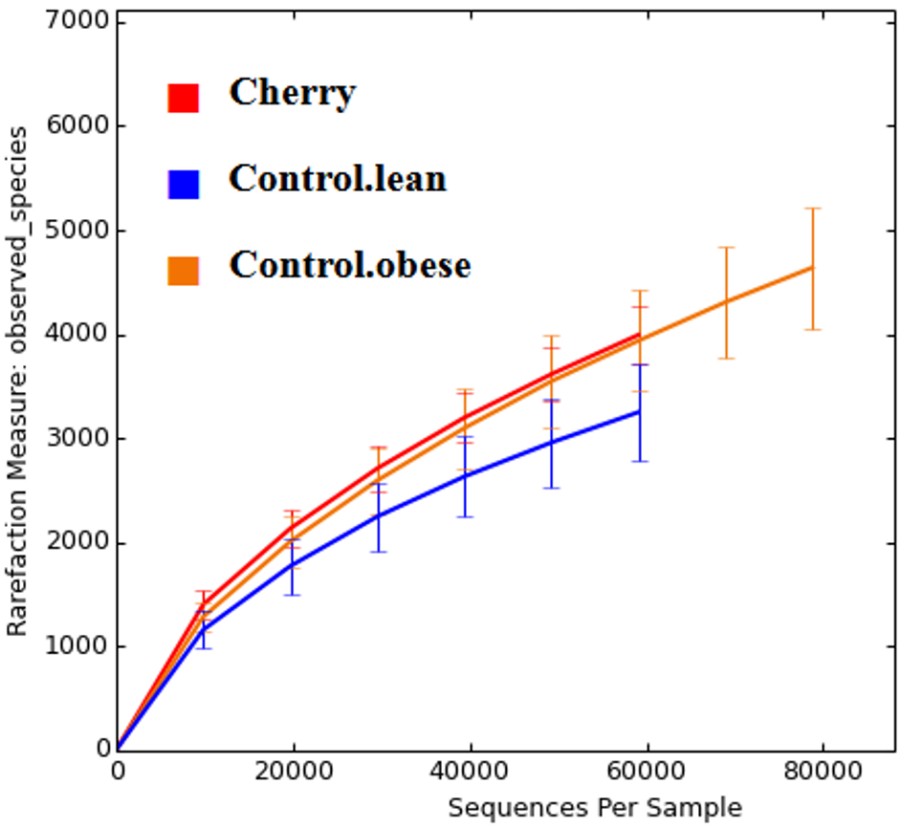

**Figure 2** **Alpha rarefaction plot for all treatment groups.** A flat line would indicate that the analysis of more sequences would not be able to detect more species (OTUs at 97% similarity).

## Alpha diversity analyses

The comparison of relative abundances help determine differences in groups of microorganisms (e.g., *Akkermansia*) but it does not help shed light into the diversity of microbial life among the different samples. Interestingly, cherry-supplemented mice showed the highest Shannon diversity indexes (index $= 8.1$) compared to both obese (index $= 7.4$) and lean controls (index $= 6.9$, $p = 0.0141$ Kruskal-Wallis). Also, the number of species (OTUs at 97% similarity) was higher in obese controls (5,407) and lower in lean (4,078), with supplemented mice having intermediate values (4,838, $p = 0.0078$, Kruskal-Wallis test) but overall the number of OTUs did not reach a plateau for any treatment group, particularly in obese mice with and without cherry (Fig. 2). This means that the sequencing effort in this study was not enough to fully describe the total number of species in our samples; however, it is important to remember that these OTU measures were obtained from an open OTU picking approach that does not discard sequences based on matching with reference database.

## Beta diversity analyses

The analysis of individual taxa such as *Akkermansia* or *Lactobacillus* yields valuable information about the membership of the bacterial communities; however, the differences

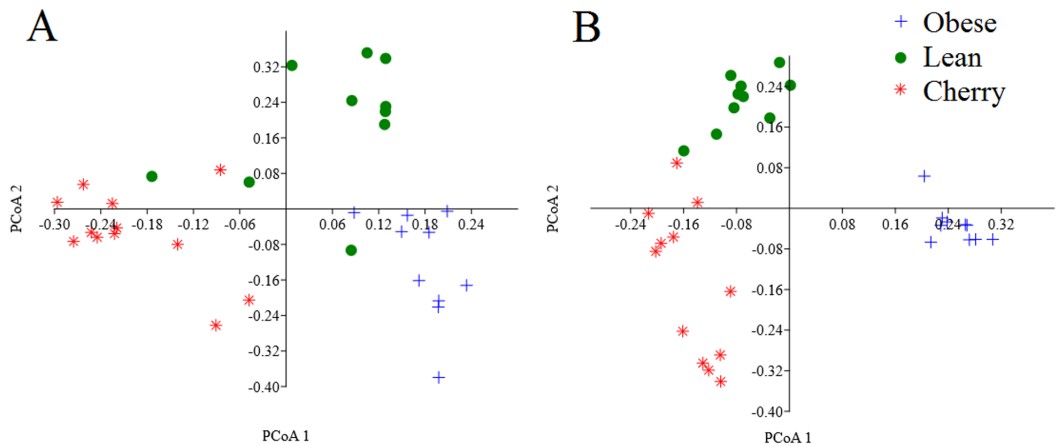

**Figure 3** **PCoA plots of weighted (A) and unweighted (B) UniFrac distance matrices.** Please note that the clustering of samples according to treatment is stronger (i.e., show less overlap) in the plot using the unweighted UniFrac distance matrix ($p = 0.001$, $R = 0.844$, ANOSIM test) compared to the plot using the weighted UniFrac distance matrix ($p = 0.001$, $R = 0.716$, ANOSIM test).

in individual taxa may or may not be sufficient to generate a distinctive microbial community. Using weighted UniFrac distances (which takes into account both phylogenetic divergence and the numbers of sequences associated with each OTU), there was a significant clustering of samples according to treatment ($p < 0.001$, Adonis test; $p = 0.001$, $R = 0.716$, ANOSIM test) (Fig. 3). There was also a significant difference using unweighted UniFrac ($p < 0.001$, Adonis test; $p = 0.001$, $R = 0.844$, ANOSIM test), which does not take into account the number of OTUs (Fig. 3). Please note that these tests often have a low sensitivity (they usually detect a difference when there is none), therefore it is also informative to look at the $R$ values in the ANOSIM test to investigate the strength of clustering (the closest to 1 the strongest the clustering of samples). Therefore, the higher $R$ value in the ANOSIM test for the unweighted UniFrac implies that the clustering is stronger compared to the clustering using weighted UniFrac, meaning that each treatment is mainly associated with phylogenetic distinct bacterial populations rather than the numbers of these populations. This result can easily be appreciated in the PCoA plots of UniFrac metrics (Fig. 3).

## Predicted metabolic profile

PICRUSt is useful at predicting the metabolic profile of the microbiota based on 16S sequencing data. In this study, a great number of features showed statistical significance among treatment groups (Table 2), especially within metabolism and genetic information processing pathways. In our experience, these differences are not due to stochastic variations (e.g., other studies from our research group have shown no differences in any feature using a very similar methodological approach, *Garcia-Mazcorro et al., 2017*). Interestingly, for almost all features cherry-supplemented mice showed abundances that were in between those abundances in obese and lean controls (Table 2). Lean mice had higher weighted Nearest Sequenced Taxon Index (weighted NSTI) scores compared to obese controls and

**Table 2 PICRUSt results (average percentages) for the statistically significant features ($p < 0.05$ adjusted $p$-values[*]).**

| Level 1 | Level 2 | Level 3 | Obese controls | Obese supplemented | Lean controls | Adjusted $p$ values |
|---|---|---|---|---|---|---|
| Metabolism | Amino acid metabolism | Amino acid related enzymes | 1.56↓ | 1.64 | 1.74↑ | 0.008 |
| Metabolism | Amino acid metabolism | Histidine metabolism | 0.61↓ | 0.72 | 0.79↑ | 0.011 |
| Metabolism | Biosynthesis of other secondary metabolites | Stilbenoid, diarylheptanoid and gingerol biosynthesis | 0.00↓ | 0.00 | 0.01↑ | 0.020 |
| Metabolism | Carbohydrate metabolism | Ascorbate and aldarate metabolism | 0.21↑ | 0.19 | 0.13↓ | 0.023 |
| Metabolism | Carbohydrate metabolism | Butanoate metabolism | 0.85↑ | 0.74 | 0.73↓ | 0.038 |
| Metabolism | Carbohydrate metabolism | Pentose and glucuronate interconversions | 0.70 | 0.73↑ | 0.62↓ | 0.049 |
| Metabolism | Carbohydrate metabolism | Pentose phosphate pathway | 0.98↓ | 1.09↑ | 1.01 | 0.033 |
| Metabolism | Energy metabolism | Carbon fixation in photosynthetic organisms | 0.68↓ | 0.74↑ | 0.73 | 0.013 |
| Metabolism | Enzyme families | Peptidases | 2.08↓ | 2.14 | 2.32↑ | 0.026 |
| Metabolism | Enzyme families | Protein kinases | 0.49↑ | 0.40 | 0.34↓ | 0.004 |
| Metabolism | Glycan biosynthesis and metabolism | Peptidoglycan biosynthesis | 0.83↓ | 0.84 | 0.94↑ | 0.039 |
| Metabolism | Lipid metabolism | Alpha-linolenic acid metabolism | 0.03↑ | 0.01 | 0.01↓ | 0.025 |
| Metabolism | Metabolism of cofactors and vitamins | One carbon pool by folate | 0.64↓ | 0.72 | 0.76↑ | 0.036 |
| Metabolism | Metabolism of cofactors and vitamins | Thiamine metabolism | 0.51↓ | 0.56 | 0.58↑ | 0.035 |
| Metabolism | Metabolism of terpenoids and polyketides | Terpenoid backbone biosynthesis | 0.54↓ | 0.61 | 0.69↑ | 0.003 |
| Metabolism | Xenobiotics biodegradation and metabolism | 1,1,1-Trichloro-2,2-bis(4-chlorophenyl)ethane (DDT) degradation | 0.00↓ | 0.00↑ | 0.00 | 0.015 |
| Genetic information processing | Replication and repair | Mismatch repair | 0.84↓ | 0.92 | 0.97↑ | 0.005 |
| Genetic information processing | Translation | Ribosome | 2.26↓ | 2.46 | 2.82↑ | 0.006 |
| Genetic information processing | Replication and repair | DNA replication proteins | 1.28↓ | 1.38 | 1.49↑ | 0.006 |

(*continued on next page*)

| Level 1 | Level 2 | Level 3 | Obese controls | Obese supplemented | Lean controls | Adjusted *p* values |
|---|---|---|---|---|---|---|
| Genetic information processing | Translation | Translation factors | 0.54↓ | 0.59 | 0.66↑ | 0.006 |
| Genetic information processing | Replication and repair | Base excision repair | 0.47↓ | 0.51 | 0.55↑ | 0.011 |
| Genetic information processing | Replication and repair | DNA repair and recombination proteins | 2.99↓ | 3.15 | 3.37↑ | 0.012 |
| Genetic information processing | Replication and repair | Nucleotide excision repair | 0.37↓ | 0.45 | 0.46↑ | 0.014 |
| Genetic information processing | Replication and repair | DNA replication | 0.69↓ | 0.74 | 0.82↑ | 0.016 |
| Genetic information processing | Translation | Aminoacyl-tRNA biosynthesis | 1.16↓ | 1.25 | 1.39↑ | 0.016 |
| Genetic information processing | Folding, sorting and degradation | Protein export | 0.61↓ | 0.65 | 0.72↑ | 0.017 |
| Genetic information processing | Transcription | RNA polymerase | 0.16↓ | 0.17 | 0.20↑ | 0.019 |
| Genetic information processing | Replication and repair | Homologous recombination | 0.95↓ | 0.99 | 1.10↑ | 0.022 |
| Environmental information processing | Signal transduction | Two-component system | 2.30↑ | 1.91 | 1.63↓ | 0.024 |
| Environmental information processing | Signaling molecules and interaction | Bacterial toxins | 0.12↓ | 0.16 | 0.16↑ | 0.025 |

**Notes.**
↓lowest.
↑highest.
*We removed five features related to human diseases that also reached statistical significance because of their questionable relevance to this study.

supplemented mice ($p < 0.05$ for both comparisons), meaning that the microbiota of all obese mice was relatively more represented in sequenced genomes.

The feature with the lowest *p* value in PICRUSt analysis was associated with terpenoid backbone biosynthesis (adjusted $p = 0.0003$), with lean controls having the highest (average: 0.69%) and obese controls the lowest (0.54%) values, with supplemented mice having values in between (0.61%). This topic is interesting because terpene synthases are widely distributed in bacteria (*Yamada et al., 2015*) and terpenes have beneficial properties in human health (*Cho et al., 2017*); however, there is little information about the potential of terpene synthesis in the gut microbiota. In order to investigate what bacterial group was more associated with this difference, we performed PICRUSt on different taxa independently using filtered OTU tables. There was a difference in this feature for the phylum Firmicutes and Bacteroidetes but all obese mice (with and without supplementation) showed very similar abundances compared to lean, suggesting that the group responsible for the overall effect on terpenoid backbone biosynthesis was not a member of either phylum. Interestingly, the independent analysis of Proteobacteria revealed that lean controls had the highest (average: 0.51%) and the obese controls the lowest values

(average: 0.39%) with supplemented mice somewhere in between (average: 0.45%), a result that is similar to the analysis of all bacterial groups at once. This suggests that a member of Proteobacteria was likely associated with the observed difference in the abundance of genes associated with terpenoid backbone biosynthesis. However, the independent analysis of individual taxa within the Proteobacteria did not yield any useful information with regards to any specific taxa associated with the overall difference in terpenoid backbone biosynthesis, suggesting that this difference was due to the combined contribution of several bacterial groups. This area is indeed worth exploring because terpenoids can work as antibiotics and growing research show that commensal microorganisms can generate potent small molecules (*Modi, Collins & Relman, 2014*). Doing this additional analysis for all features that showed statistical significance (Table 2) is advisable but is outside of the scope of this present manuscript.

## qPCR analyses

In this study we performed qPCR analyses using DNA from both colon contents and mucosal samples. Using DNA obtained from samples of colon contents, several qPCR results confirmed the sequencing results (Fig. 4). For example, *Lactobacillus* was found to be lower in cherry-supplemented mice compared to both lean and obese controls ($p < 0.0001$). Similarly, *Akkermansia* was lower in obese mice compared to lean and cherry supplemented ($p < 0.0001$, Kruskal Wallis test) and this result was mainly due to a difference between the obese control group and the cherry-supplemented group ($p < 0.0001$) (Fig. 4). Also, Enterobacteriaceae were lower in cherry-supplemented and lean mice compared to obese controls. Using qPCR we were able to show that *E. coli* was also lower in supplemented mice, a finding that we could not investigate using sequencing. Moreover, Betaproteobacteria was higher in cherry-supplemented compared to lean and obese controls ($p < 0.0001$). Please note that the family Alcaligenaceae within the Betaproteobacteria also showed similar differences using sequencing. qPCR results for colon contents also showed that the abundance of *Bifidobacterium* was higher in supplemented mice compared to obese ($p < 0.0001$) and lean ($p = 0.002$) controls (Fig. 4). These results were partly confirmed using sequencing at higher taxonomic levels (order Coriobacteriales, see Colonic microbiota above). *C. butyricum* (a butyrate-producing microorganism, *Zhang et al., 2009*) was also found to be higher in supplemented mice (Fig. 4).

qPCR analysis of the mucus-associated microbiota helped shed light into an area that is not usually evaluated (most studies evaluate either feces or intestinal contents in part due to ease of sampling and amount of material for analysis). Unfortunately, we only obtained results from four bacterial groups because the results from all other bacterial groups (*Akkermansia* included) were either undetectable or fell below the lowest standard. This can be explained by the fact that commensal bacteria have their habitat in the outer colonic mucus layer, which can be easily lost during tissue dissection and washing (*Johansson et al., 2008*). However, opportunistic pathogens have developed mechanisms to secrete proteases that cleave mucin allowing certain bacteria penetrate and reside in the inner mucus layer (*Pelaseyed et al., 2014*). Interestingly, in this study we found patterns of variations in colon mucosal cells that were not in agreement with those variations observed in colon contents.

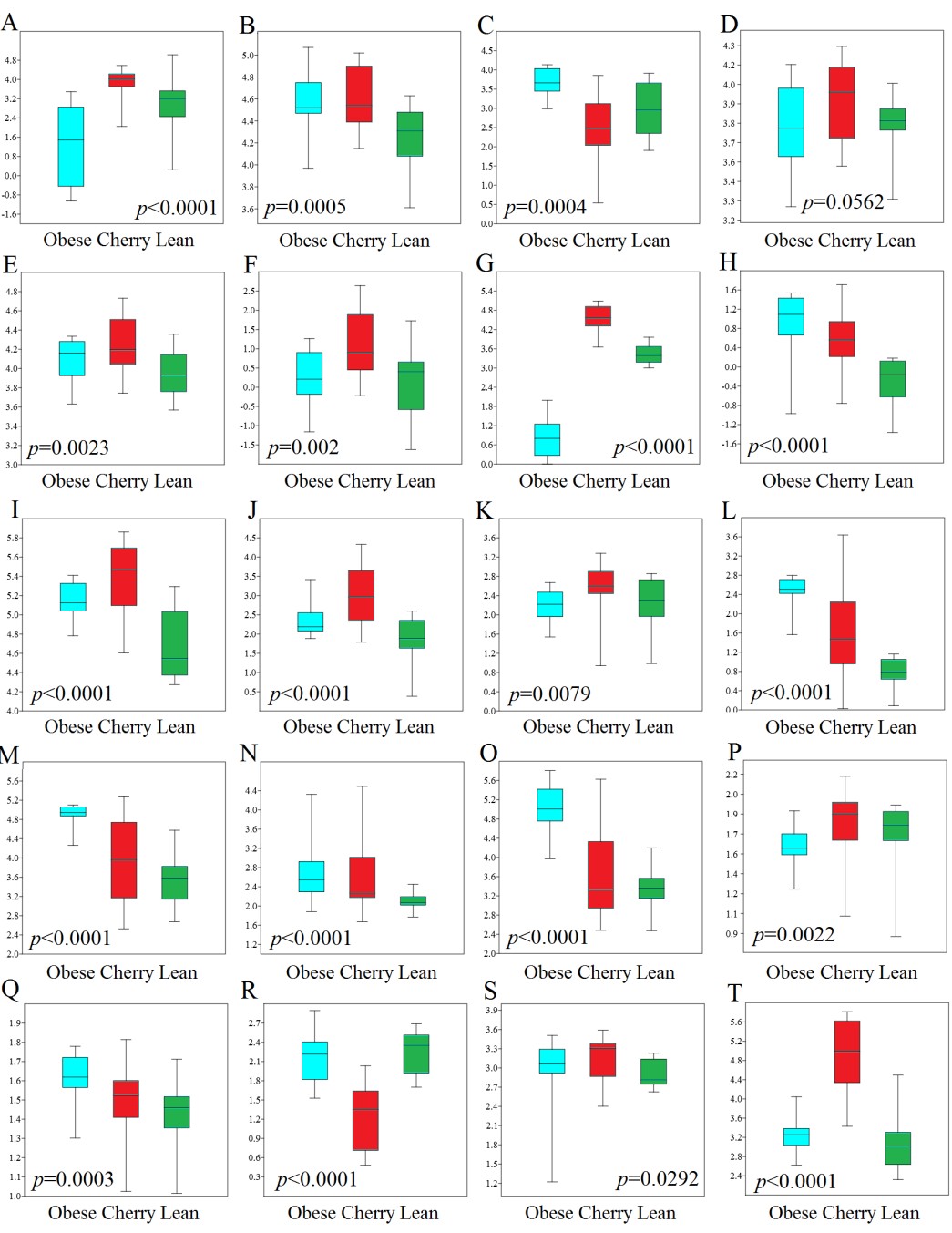

**Figure 4** **Boxplots showing qPCR results for selected bacterial groups in colon contents (or colon mucosa) that showed or almost reached statistical significance difference.** *P* values come from the Kruskal-Wallis test. (A) *Akkermansia*, (B) *Bacteroides fragilis*, (C) *Bacteroides vulgatus*, (D) *Bacteroides/Prevotella*, (E) Bacteroidetes, (F) Bacteroidetes (colonic mucosa), (G) Betaproteobacteria, (H) Betaproteobacteria (colonic mucosa), (I) *Clostridium butyricum*, (J) *Clostridium butyricum* (colonic mucosa), (K) *Clostridium* cluster IV, (L) *E. coli*, (M) Enterobacteriaceae, (N) Enterobacteriaceae (colonic mucosa), (O) *Enterococcus*, (P) *Faecalibacterium*, (Q) *Lactobacillus plantarum*, (R) *Lactobacillus*, (S) Ruminococcaceae, (T) Tenericutes. qPCR data is expressed as log amount of amplified DNA (in picograms) per 10 ng of total isolated DNA.

**Table 3** **Median (minimum–maximum) SCFA concentrations ($\mu$mol/mg caecal contents).** *P* values come from either the Kruskal–Wallis test or the Mann Whitney test when comparing only two treatment groups due to lack of detectable values in one group. Different letters state statistical significance difference. The symbol (−) is included to denote treatment groups where all or most samples were undetectable. The number of samples (*n*) in which the specific SCFA was detected for each experimental group is also included. For most SCFAs, we chose not to perform a statistical comparison because of very low sample size in at least one treatment group (NA or not applicable). Please note that most samples (especially from lean and obese controls) showed undetectable levels of several SCFAs. In our experience this was not due to errors in our analytical methodology, and this is supported by the fact that all samples were treated equally yet most samples from supplemented mice did show detectable levels of most SCFAs.

| | Obese controls | Obese supplemented | Lean controls | *P* value |
|---|---|---|---|---|
| Caproate | 1.2(0.4–3.9)[a] (*n* = 9) | 285(217–437)[b] (*n* = 12) | 1.0(0.4–652)[a] (*n* = 10) | 0.0033 |
| Methyl butyrate | – | 116(17–405) (*n* = 12) | 62(43–92) (*n* = 3) | NA |
| Butyrate | 6.2(5.3–20)[a] (*n* = 9) | – | 11.9(6.1–16.2)[a] (*n* = 7) | 0.3511 |
| Propionate | – | 384(258–649) (*n* = 12) | 356(281–438)(*n* = 4) | NA |
| Acetate | 1.9(1.4–1.9) (*n* = 3) | 269.4(128–672) (*n* = 12) | 273.2(40–351) (*n* = 3) | NA |
| Valerate | – | 15.4(4–48) (*n* = 10) | – | NA |

For example, Betaproteobacteria and Enterobacteriaceae were found to be lower in lean mice compared to all obese mice with and without cherry supplementation (Fig. 4). On the other hand and similarly to qPCR results in colon contents, *C. butyricum* was also higher in supplemented mice and Bacteroidetes also showed similar results compared to qPCR results from colon contents (Fig. 4).

## SCFA in caecal contents

SCFA are microbial metabolites that have been associated with health and disease. In this study, the SCFAs detected in cherry-supplemented mice (caproate, methyl butyrate, propionate, acetate and valerate) exceeded those concentrations detected in obese and lean controls except for butyrate, either because of higher values or because of higher number of samples in which the SCFAs were detected (Table 3). This is relevant because SCFAs are substrates for colonocytes providing at least 60–70% of their energy requirements (*Suzuki, Yoshida & Hara, 2008*) with implications for gut barrier function. Based on these results, cherry dietary supplementation contributes with fiber and phytonutrients that apparently promotes a healthier SCFA-producing microbiota, thus contributing to improve colon barrier function and reduce the risk of inflammatory diseases (*Tan et al., 2014*).

## Outer colon wall thickness

In this study, the median ratio outer colon wall to total colon wall was higher in supplemented mice (median = 0.73) and lean mice (median = 0.72) compared to obese controls (median = 0.64) but this difference did not reach significance ($p = 0.08$) due to the variability among animal subjects (Fig. S1). Increased intestinal concentrations of

SCFAs in supplemented mice might have contributed to increase the height of intestinal outer wall layer (*Ramos et al., 1997*).

### Biomarkers of inflammation, cellular stress, and gut barrier function in colon tissue and colonic mucosal cells

The colon mucus layers and enterocytes provide the first defense line of the gastrointestinal tract. We have analyzed the mRNA levels of biomarkers of inflammation, cellular stress, and gut barrier function in mucosal cells and colon tissues as a tool to assess whether the changes promoted in gut microbiota by cherry bioactive compounds might also trigger differentiated responses in the host gate keepers (mucosal layer and epithelial cells) with possible implications in host-bacterial interactions and host immune system. Despite the differences in microbiota and SCFAs, most of the biomarkers analyzed in colonic mucosal cells were similar between experimental groups (Table S5), and no difference was found in biomarkers assessed in colon tissues (Table S6). In colonic mucosal cells only ATF4 mRNA levels were significantly lower in supplemented group compared to lean ($p < 0.05$), and tended to be lower than in obese control. ATF4 is a stress-induced transcription factor whose expression has been correlated with degree of intestinal inflammation and development of inflammatory bowel diseases in adults (*Negroni et al., 2014*). Likewise, VCAM-1, known to control leukocyte/ monocyte intestinal recruitment and localization in LPS-induced inflammation (*Totsuka et al., 2014*), were lower in supplemented group than obese and lean controls, but did not reach significance ($p = 0.06$) (Table S5).

The concentrations of LPS measured as EU in caecal contents and blood plasma showed no significant difference among experimental groups. However, the LPS concentrations in lumen were not determined due to limitations in sample availability used for DNA extractions and microbiota analysis. Thus, LPS in caecal contents might not necessarily correlate with the LPS concentrations in lumen. We could speculate LPS was lower in supplemented group, thus explaining the lower ATF4 mRNA levels.

### Correlation analysis

The separate analyses of microbiota, SCFAs and mRNA levels yielded useful information with regard to the effect of dietary interventions but this data must be integrated in an effort to find possible biologically relevant associations. Overall, data from 16S rRNA sequencing was highly correlated with data from qPCR (Fig. S2), as discussed above. Interestingly, a high positive correlation was observed between *Bacteroides* and *Akkermansia*, while a negative correlation between *Bacteroides* and *Lactobacillus* was observed. It is also worth mentioning that samples from supplemented mice were associated with higher levels of SCFAs and more *Akkermansia*, an interesting relationship given the production of SCFAs by this bacterial group (*Belzer & De Vos, 2016*).

### DISCUSSION

Diabetes and obesity are complex diseases that can often be treated using a combination of medications, dietary modifications and physical exercise. Cherries contain fiber and bioactive compounds such as polyphenolics that can promote wellbeing in the host. This

study describes the effect of cherry consumption on the colonic microbiota, short-chain fatty acids, and biomarkers of intestinal health using an *in vivo* model of genetic obesity.

The metabolism and pharmacokinetics of cherry bioactive compounds inside the host are important to evaluate any possible effect of a dietary intervention with cherry. A recent study showed evidence suggesting an involvement of glucose transporters in the small intestine (such as the sodium-dependent linked transporter) in the absorption of anthocyanins from bilberries but it also highlighted the wide differences in bioavailability among different types of anthocyanins (*Baron et al., 2017*). The amount and chemical characteristics of any post-digestion bioactive compounds that reach the large intestine also varies depending on several factors. Importantly, these bioactive compounds are often transformed throughout the digestive tract and reach the lower intestine in a modified form *Stalmach et al. (2010)*. It has been shown that the amount of material reaching the colon is considerable and some authors even catalogue some of these compounds as prebiotics because of its effect in the abundance of certain microorganisms (*Cires et al., 2017*). Please note that prebiotics are historically considered to be non-digestible fiber and that the increase in abundance of a certain group of microorganisms (e.g., *Lactobacillus*) when exposed to anti-oxidants may or may not involve direct feeding on the compounds such as in the case of dietary fiber. This is further complicated in case of cherries which contain both fiber and considerable amounts of polyphenolics (*McCune et al., 2011*; *Wang et al., 2017*).

This study showed strong evidence that cherry supplementation can modify the colon microbiota, a phenomenon that may be related to the fiber and/or to any post-digestion bioactive compounds reaching the lower intestinal tract. For example, this study showed that the levels of *Akkermansia* spp. in colon contents were higher in supplemented mice compared to both obese and lean controls. *Akkermansia* is a common and relatively abundant (~1%) anaerobic member of the gut microbiota (*Derrien et al., 2008*) that is supposedly highly specialized in host-compounds that may not compete with the microbiota in the highly populated lumen and therefore do not depend on nutrients from host food consumption (*Derrien et al., 2011*). *Akkermansia* indeed deserves attention because of its potential role as mediator of improved inflammatory and metabolic phenotype of mice (*Caesar et al., 2015*). Interestingly, it has been shown that the abundance of *Akkermansia* is lower in the intestinal epithelium of patients with Inflammatory Bowel Disease (*Png et al., 2010*) and in feces often negatively correlates with body weight in rodents and humans (the higher the body weight the lower the abundance of *Akkermansia*, *Everard et al., 2013*). Another study showed that human subjects with higher *A. muciniphila* abundance in feces exhibited the healthiest metabolic status (*Dao et al., 2016*). Therefore, members of this taxon have been suggested as biomarkers for a healthy intestine (*Png et al., 2010*; *Swidsinski et al., 2011*). Accordingly, in this study we showed that cherry supplementation was associated with increases the abundance of this health-bearing microorganism, suggesting a beneficial effect of cherry consumption on health. Other similar studies from our research group have also showed that obese mice have less *Akkermansia* compared to lean and quinoa-supplemented obese mice (*Garcia-Mazcorro, Mills & Noratto, 2016a*), and others have shown that *Akkermansia* is higher during prebiotic administration (*Everard et al., 2011*; *Van den Abbeele et al., 2011*).

The reasons behind any increase or decrease of bacterial groups in colon contents are often difficult to clarify. While there may be several explanations for this phenomenon, in the case of *Akkermansia* it has been shown showed that the accompanying microbiota composition determines the magnitude and pattern of host-compounds foraging by this group (*Berry et al., 2013*), an interesting phenomenon that has also been shown in other bacteria such as *Bifidobacterium* (*Klaassens et al., 2009*). Importantly, it has been shown that *Akkermansia* is actually composed by at least eight different species based on the 16S rRNA gene (*Van Passel et al., 2011*). This heterogeneity may explain why in other studies the abundance of *Akkermansia* was not necessarily related with health status (*Garcia-Mazcorro et al., 2016b*; *Noratto et al., 2014*). In fact, *Akkermansia* was shown to be increased in mouse studies of dextran sodium sulfate (DSS)-induced colitis (*Berry et al., 2012*; *Kang et al., 2013*; *Håkansson et al., 2015*) and can seemingly aggravate *Salmonella enterica* Typhimurium-induced gut inflammation in a gnotobiotic mouse model (*Ganesh et al., 2013*). This phenomenon has been explained by an outgrowth of *Akkermansia* in response to the thickening of the mucus layer (*Ottman et al., 2017*) but it fails to explain the rise of this group in intestinal contents. Given the potential existence of different species of *Akkermansia* (*Van Passel et al., 2011*), it is fair to speculate that at least some of these species can use dietary substrates (instead or in addition to host-compounds) and proliferate in the intestinal lumen, thus explaining the rise in numbers. On the other hand, feces also contain abundant mucus (*Swidsinski et al., 2008*), therefore it is also possible that the overgrowth of *Akkermansia* in the mucus layer simply leaked into the lumen.

Other bacterial groups aside *Akkermansia* deserve attention. For example, this study also showed that a group of fiber degraders (the S24-7 family) was also higher in the supplemented group, a finding likely related to the fiber contributed by the cherry-supplemented diet. The genomes of several members of this S24-7 family have been recently explored showing that it contains three trophic guilds, each broadly defined by differential abundances of enzymes involved in the degradation of specific carbohydrates (plant, host and α–glucan) (*Ormerod et al., 2016*). Interestingly, cherry consumption was also associated with lower levels of *Lactobacillus* (a commonly health-bearing group of bacteria) and Enterobacteriaceae (a group comprising potential pathogens). While the reason behind these changes are likely related to the complex microbial-host milieu, a recent study showed that these two groups (both *Lactobacillus* and Enterobacteriaceae) were positively correlated to high levels of *Salmonella*-induced inflammation (*Borton et al., 2017*).

This study showed that the differences in bacterial populations in the supplemented group were accompanied by concentrations of SCFAs that generally exceeded those concentrations in lean and obese controls except for butyrate, an interesting finding because it is precisely butyrate that has caught more attention from the scientific community. Butyrate is produced and also transformed by the gut microbiota; however in our study the increased concentrations of methyl butyrate (a methyl ester of butyric acid) in the supplemented group might be associated with the presence of this ester in cherries as occurs in many plant products. This might be an advantage over production of butyrate because the latter is rapidly metabolized and has limited clinical efficacy in contrast with methyl butyrate which is less polar and less susceptible to being cleared by the body (*Khan,*

*Ahmad & Srivastava, 2016*). On the other hand, methyl butyrate may also be produced as a by-product of bacterial metabolism. For example, in one study high production of methyl butyrate was the most significant change induced upon prebiotic and synbiotic supplementation in fecal fermentation *in vitro* (*Vitali et al., 2012*). Here it is also important to highlight the difficulties at determining which bacterial group contributes to each SCFA because multiple groups are often involved in the production/degradation of chemical compounds in the gut (*Pryde et al., 2002*).

Here we show that cherry bioactive compounds (including fiber) modifies the colonic microbiota and SCFA in obese diabetic mice and we hypothesized that this modification may trigger changes in the gut immunity and physiology of the host. However, here we also showed that cherry powder supplementation did not affect mRNA expression of several biomarkers associated with gut health in spite of a change in bacterial composition and biochemistry. Only ATF4 mRNA levels were different among experimental groups and downregulated in cherry-supplemented mice. ATF4 can be upregulated in response to bacterial LPS as adaptive response that triggers the expression of inflammatory cytokines (TNF-α, IL-6, and IL-10), signaling pathways (NF-kB and MAPK), and the ATF4-CHOP apoptotic pathway (*Rao et al., 2013*). Therefore, downregulation of ATF4 in supplemented group might be linked to lower levels of bacteria producing LPS in the colon mucus layer, with implications for intestinal cells inflammation and survival. The LPS concentrations in caecal contents might not resemble LPS in lumen layer in contact with the colon mucus layers. Unfortunately, this could not be confirmed due to the limitations intrinsic to the collection of specific fractions of lumen for LPS determination.

In addition, the lower levels of ATF4 and VCAM-1 may also be mediated by the higher SCFAs production in supplemented mice as demonstrated by *Huang et al. (2017) in vitro*. SCFAs inhibited oxidative stress, inflammatory response, and cell adhesion molecules induced by LPS and glucose through activation of their specific G protein-coupled receptors 43 (GPR43) (*Huang et al., 2017*). Even though our experimental conditions did not compromise gut integrity and barrier function as confirmed by LPS in plasma that was similar among experimental groups, our study has provided insights to future studies investigating cherry intake within the context of acute and chronic intestinal inflammation. The improvement in intestinal integrity by polyphenolics enriched extracts, independently of alterations in gut microbiota, was demonstrated over a period of only four weeks in pigs induced subclinical chronic inflammation with *E. coli* LPS injections (*Liehr et al., 2017*). Because the observed changes in bacterial composition and SCFAs could have been related to bioconversion of cherry compounds, it would be interesting to study the effect of cherry compounds in germ-free mice, with the obvious disadvantage of not representing a real-life scenario. In general, our results reveal interesting research avenues for cherry intake within the context of chronic and acute intestinal inflammation using conventional and germ-free mice to determine whether the cherry-induced intestinal bacteria modulation could be beneficial in ameliorating or preventing the symptoms of intestinal inflammation.

This study has drawbacks that ought to be taken into account in future studies. First, cherries contain fiber and a wide variety of bioactive compounds and this is important because not all compounds have the same properties on health. In this regard it was not the

objective of this study to identify each compound individually but to assess the effect of the fruit as a whole, at least in a powder form. Second, in this study we used a model of genetic obesity but diet-induced obesity can also shed light into the mechanisms associated with any health effect (*Hariri & Thibault, 2010*). One main disadvantage of using diet-induced obesity is the high number of diets that can be used to promote obesity and perhaps more importantly the nutritional differences and outcomes in host physiology among these diets. Also, we know now that obesity in humans is a complex disorder that often involves a genetic difference in the host. Importantly, *Song et al. (2016)* showed a beneficial effect of cherries using a diet-induced model of obesity, thus suggesting that cherries can have a positive impact in different types of obesity-related disorders.

## CONCLUSIONS

In summary, this study shows that cherry supplementation for 12 weeks can modify the colon microbiota and the concentrations of SCFAs. Our research model did not provide strong evidence to suggest that this dietary intervention can lead to changes in biomarkers of inflammation, cellular stress, and gut barrier function in colonic mucosal cells and colon tissues. The reason of why the change in the microbiota and SCFAs did not affect the host physiology remains to be investigated, but may be related to the obese genetic animal model used instead of using high fat, high sugar to induce obesity, which is known to stimulate intestinal inflammation (*Rahman et al., 2016*). Also, it has been shown that mice deficient for intestinal gluconeogenesis do not show the same metabolic benefits on body weight and glucose control induced by SCFAs, despite similar modifications in gut microbiota composition (*De Vadder et al., 2014*). Indeed, this topic is worth exploring further, especially in a context of diabetes (*Mithieux et al., 2004*). More research is desirable into the implications of cherry consumption as a dietary supplement in diabetic and obese human patients.

## ACKNOWLEDGEMENTS

We thank Mary J. Thompson for her technical assistance in mice handling and Mohit Sharma for his assistance in colon data analysis.

### Funding

This work was supported by the Washington State Department of Agriculture and the Washington State Fruit Commission through SCBGP Grant K-1263. The funders had no role in study design, data collection and analysis, decision to publish, or preparation of the manuscript.

### Grant Disclosures

The following grant information was disclosed by the authors:
Washington State Department of Agriculture.
SCBGP Grant: K-1263.

## Competing Interests

Jose F. Garcia-Mazcorro is an employee of MNA de Mexico, San Nicolas de los Garza, Nuevo Leon, Mexico and Scot Dowd is an employee of Molecular Research LP, Shallowater, Texas, USA.

## Author Contributions

- Jose F. Garcia-Mazcorro and Nara N. Lage performed the experiments, analyzed the data, wrote the paper, prepared figures and/or tables, reviewed drafts of the paper.
- Susanne Mertens-Talcott and Stephen Talcott conceived and designed the experiments, analyzed the data, contributed reagents/materials/analysis tools, reviewed drafts of the paper.
- Boon Chew conceived and designed the experiments, analyzed the data, contributed reagents/materials/analysis tools, wrote the paper, reviewed drafts of the paper.
- Scot E. Dowd performed the experiments, analyzed the data, reviewed drafts of the paper.
- Jorge R. Kawas analyzed the data, contributed reagents/materials/analysis tools, wrote the paper, prepared figures and/or tables, reviewed drafts of the paper.
- Giuliana D. Noratto conceived and designed the experiments, performed the experiments, analyzed the data, prepared figures and/or tables, reviewed drafts of the paper.

## Animal Ethics

The following information was supplied relating to ethical approvals (i.e., approving body and any reference numbers):

The experimental analyses carried out in this manuscript were approved by the Institutional Animal Care and Use Committee at Texas A&M University (IACUC 2013-0149).

## Data Availability

NCBI: SRP117747.

## Supplemental Information

Supplemental information for this article can be found online at http://dx.doi.org/10.7717/peerj.4195#supplemental-information.

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
