# Peer review of "Effect of dark sweet cherry powder consumption on the gut microbiota, short-chain fatty acids, and biomarkers of gut health in obese db/db mice"

_PeerJ, doi:10.7717/peerj.4195_

## Round 0.1 · original submission · Major Revisions

· Academic Editor

Major Revisions

The paper has interesting results and can be considered for publication after the revisions required by the reviewers are performed.

·

Basic reporting

Nothing to report here

Experimental design

Were the lean mice db/db as well, or another strain? It may be good to fully describe the lines used (db/db is not the complete descriptor)...
L186 colon mucosal cells normally implies host cells, which would not be Bacteria! Maybe you mean mucosal samples?

Validity of the findings

Figure 1: I am not convinced that showing results at the order level is acceptable. It is way more common to present phylum and genus (family) level.
Figure 2 has no place has a main figure, this is not "valuable" data
Figure 3: The ANOSIM should give you a p-value which is needed to conclude on the clustering!
Figure 4: It would be so much easier if you had the taxon name on each plot! H, L, J, O appear to show very similar values, which raises questions about the statistics...
Table 2: I personally have doubts about the validity of Picrust, and seems like the differences you report are all extremely slight. Without a way to test if such differences are not due to stochastic variations in the sampling, I would not report them.
Table 3: These results seem completely flawed. There is no way butyrate and propionate are not produced by gut microbes! And the 100X more values in the obese supplemented mice don't make any sense

Reviewer 2 ·

Basic reporting

Discussion could be implemented (see general comments to the author).

Experimental design

Seems quite adapted to the question raised. Studies seem competently performed.

Validity of the findings

The findings are interesting and seem valid and =adequately interpreted.

Additional comments

This is an interesting study, which questions the "power" of the gut microbiota to change metabolism of the host. Basically, from a healthy dieting (cherry powder, fiber-enriched), inducing a healthy microbiota composition (enriched in "beneficial" bacteria), and producing "beneficial" metabolites deriving from fermentation of soluble fibers (short-chain fatty acids), no substantial change in several important gut health parameters of the host was observed in a mouse model of diabetes (db/db mice). This is an important set of results, suggesting that the power of microbiota to significantly alter the host metabolism is limited by the genetic bakground of the host. This study resonates in me with another study in relation with the role of fiber-derived SCFAs to improve host glucose control by activating intestinal gluconeogenesis, the benefits being absent in intestinal gluconeogenesis-deficient mice despite the presence of SCFAs and the modification of microbiota (De Vadder et al, Cell, 2014). This suggests that host genetics may dominate the microbiota composition as for regards host metabolism. The paper would be greatly improved by inclusion of a paragraph od discussion addressing this point

---

## Round 0.2 · accepted · Accept

· Academic Editor

Accept

The manuscript has been revised as requested by the reviewers.